# Efficient Class-Incremental Segmentation Learning via Expanding Visual Transformers

## Abstract

Incrementally learning new semantic concepts while retaining existing information is fundamental for several real-world applications. Although the impact of backbone size and architectural choices has been extensively studied in non-incremental computer vision tasks for efficiency concerns, class-incremental semantic segmentation models have so far focused primarily on large backbones, without offering a fair comparison in terms of model size. In this work, we propose a fairer study across existing class-incremental semantic segmentation methods, focusing on the models efficiency with respect to their memory footprint. Moreover, we propose TILES (Transformer-based Incremental Learning for Expanding Segmenter), a novel approach exploiting small-size ViT backbones efficiency to offer an alternative solution where severe memory constraints are applied. It is based on expanding the architecture with the increments, allowing to learn new tasks while retaining old knowledge within a limited memory footprint. Besides, in order to tackle the background semantic shift, we apply adaptive losses specific to the incremental branches, while balancing old and new knowledge. Furthermore, we exploit the confidence of each incremental task to propose an efficient branch merging strategy. TILES outperforms several previous methods on various challenging benchmarks while using up to 14 times fewer parameters.

## 1 Introduction

In a traditional non-incremental learning context, machine learning models process the whole training data at once. However, incrementally learning new information held by new collected data while retaining past knowledge is a critical capacity needed in real-word applications such as robotics, self-driving vehicles or video surveillance, as past data is not always available for storage or privacy reasons. In a more specific context, Class-Incremental (CI) learning can be more constrained when the new data features new classes not seen previously without bringing anymore knowledge about previously learnt classes. While humans can learn to recognize new objects continuously without forgetting the ones previously seen, deep learning models can suffer from performance degradation on previously learned tasks. This phenomenon introduced by Mccloskey & Cohen (1989) is known as *catastrophic forgetting*, and happens especially if the new classes are learned with no supervision on past knowledge.

This problem has received much attention from the research community for the task of image classification (IC). Several approaches have proposed to expand the model architecture to learn new tasks such as Yan et al. (2021) or to retain a portion of the previous dataset and use it in subsequent steps. Other researchers have tried to solve CI learning

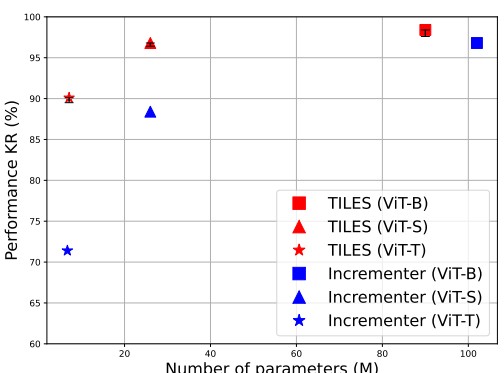

Figure 1: Model size vs. performance (Knowledge Remaining) for TILES and Incrementer with 3 different backbones on the Pascal-VOC 15-5 *overlapped* scenario. Error bars represent the variations of TILES through iterations.

in a more challenging setting, where no old data is available and with no or limited model expansion. Approaches based on transferring information from the old to the new network known as *Knowledge Distillation* (KD) presented by Hinton et al. (2015) have shown great success in alleviating catastrophic forgetting for IC. Architecture-wise, all first methods were CNN based until DyTox (Douillard et al. (2022)) which demonstrated good performances while using a Visual Transformer (ViT) based architecture (Dosovitskiy et al. (2021)) with very few additional parameters at each step.

However, less attention has been given to Class-Incremental Semantic Segmentation (CI-SS). This task holds one more challenge which is the *semantic shift of the background*: the semantic meaning of the background changes at each step since past and future classes are considered as *background* relative to the current step (which only relies on labels of current classes). This is why direct adaptations of CI-IC approaches do not perform well for CI-SS. Several solutions have been proposed to deal with this challenge such as self-inpainting by Cermelli et al. (2020). Besides, while most approaches dealing with CI-SS use Deeplab-v3 (Chen et al. (2017)) with a CNN backbone, more recently, CoinSeg (Zhang et al. (2023)), NeST (Xie et al. (2024)) and Incrementer (Shang et al. (2023)) leveraged ViT-based architectures and outperformed CNN-based models while employing large ViT backbones. However, to the best of our knowledge, no benchmark exists to fairly compare these various methods. Moreover, despite the various studies providing smaller sized ViT backbones on several computer vision tasks (Strudel et al. (2021); Liu et al. (2021)), no such small models have been proposed for CI-SS.

In this work, we provide a fairer comparison across previous state-of-the-art (SOTA) CI-SS approaches, by taking into account the performance of the used backbones having different memory footprints, disentangling the improvements attributable to the architectural design from those resulting from the incremental learning strategy. Besides, we demonstrate that previous ViT-based CI-SS methods using large backbones are not adapted for use cases with severe memory constraints where smaller backbones should be used. As an alternative, we propose a novel approach to solve limited footprint CI-SS based on an expanding ViT architecture. Similarly to Incrementer, we choose Segmenter architecture (Strudel et al. (2021)) for its class embeddings which represents an efficient way to retain knowledge on classes. In contrast, our method features an expanding architecture which helps alleviate the semantic shift of the background, especially when using small backbones. Therefore, our Transformer-based Incremental Learning for Expanding Segmenter (TILES) appears better suited to the constrained-memory CI-SS scenarios as it largely outperforms previous SOTA methods when limited-sized backbones are applied (see Figure 1 for an illustration). It is also able to outperform several previous approaches while using up to 14 times fewer parameters. The effectiveness of the proposed framework is demonstrated through extensive experiments on the CI-SS benchmarks Pascal-VOC (Everingham et al. (2010)) and ADE20k (Zhou et al. (2019)). Our contributions can be formulated as follows:

- We provide a fair comparison across different CI-SS methods while studying their behaviors and efficiencies. Besides, we demonstrate a big performance drop of the best SOTA approach when using smaller backbones.

- Alternatively, we propose TILES to solve CI-SS with critical memory constraints. We demonstrate that the proposed expanding mechanism dedicating balanced task-specialized branches alongside a suitable loss computation and a branch merging technique, are the more efficient to cope with constrained-memory cases.

- We improve the SOTA performance for CI-SS using limited-sized backbones, on two challenging benchmarks.

## 2 Related work

### 2.1 Class-incremental learning

We can distinguish three main families of approaches solving CI learning for image classification, object detection or semantic segmentation:

**Replay methods:** The strategy here is to use data holding past knowledge in the subsequent steps. This can be done either by saving images or feature samples from the previous datasets at each step (Rebuffi et al. (2017); Lopez-Paz & Ranzato (2017); Pellegrini et al. (2020)), by training a generative model to generate additional training samples similar to previous datasets (Shin et al. (2017); Liu et al. (2020)), or by using web scraping to collect samples from the internet which will be added to the new dataset (Maracani et al. (2021)). This idea can provide good performances since there is limited or no catastrophic forgetting, but it is contrary to privacy constraints (*e.g* health, industrial or defense sensitive data). In this work, we do not consider this family of approaches.

**Expansion methods:** These approaches try to find an efficient way to make the model *evolve* throughout the epochs by allocating new parameters to the new classes. This strategy has already proven its worth in CI-IC by achieving good performance in Yan et al. (2021). DyTox (Douillard et al. (2022)) is the first approach to use an expanding transformer-based architecture with limited sized backbones for CI-IC. It can dynamically process new knowledge by moderately increasing the number of parameters leading to performance similar to SOTA CNN-based approaches.

**Regularization methods:** This family of approaches focuses on how the model learns at each step and can be further divided into weight regularization and functional-based methods. On the one hand, *weight regularization* approaches, especially used for CI-IC, put constraints on the weights having high impact on previous tasks (Kirkpatrick et al. (2017)). On the other hand, *functional-based* approaches compute losses measuring the distance between a specific layer outputs produced by the previous and the new models respectively (Cermelli et al. (2020); Shmelkov et al. (2017); Phan et al. (2022)). These methods are the most popular thanks to their simplicity and capacity to continuously learn new classes. In particular, *Knowledge Distillation* (KD) proposed by Hinton et al. (2015) has shown great success reducing the catastrophic forgetting especially for CI-IC (Rebuffi et al. (2017)).

## 2.2 Class-incremental semantic segmentation

Several methods have tried to solve CI-SS using KD, given its success for CI-IC. However, the Cross Entropy Loss ($L_{CE}$) and the KD Loss ($L_{KD}$) contradict each other for CI-SS because of the semantic shift of the background. In fact, previously seen classes are labeled in the new ground-truth as background, hence a contradiction with the old model's predictions. Similarly, the new classes are classified as background by the old model, hence a contradiction with the new ground-truth labels. Therefore, this opposition makes the model cut the trade-off between rigidity and flexibility in an inefficient way.

MiB (Cermelli et al. (2020)) is the first to tackle the background semantic shift challenge by changing the new model outputs depending on which loss is computed. Indeed, the new model predictions corresponding to new classes are considered as background in the $L_{KD}$. Similarly, those corresponding to old classes are considered as background in the $L_{CE}$. PLOP (Douillard et al. (2021)) deals with the background semantic shift in a different manner, by joining the predictions made by the previous model and the new ground truth to generate a new target. Later, SDR (Michieli & Zanuttigh (2021)) and UCD (Yang et al. (2022)) show that using contrastive learning on lower dimension representations offers a powerful way to retain knowledge. RCIL (Zhang et al. (2022a)) uses representation compensation to merge old and new parameters while MicroSeg (Zhang et al. (2022b)) addresses the challenge of background shift by leveraging regional objectness to identify and preserve previously learned classes. Moreover, RBC (Zhao et al. (2022)) corrects context bias through a context-rectified image-duplet learning scheme, a biased-context-insensitive consistency loss, and an adaptive class-balanced learning strategy. Finally, Bg_Adapt (Zhang & Gao (2024)) leverages a background adaptation mechanism based on residual modeling to better handle background category evolution. Even though these methods propose different incremental strategies, they are all based on the Deeplab-v3 (Chen et al. (2017)) architecture with a ResNet (He et al. (2016)) backbone as the core model on which the CI-SS strategies are applied. More recently, two methods supporting both semantic and panoptic segmentation adopt the Mask2Former Cheng et al. (2022) architecture while always considering a ResNet backbone. In fact, ComFormer (Cermelli et al. (2023)) frames segmentation as a mask-classification task, and overcomes catastrophic forgetting through a novel adaptive distillation loss combined with mask-based pseudo-labeling. ECLIPSE (Kim et al. (2024)) freezes the base model and incrementally fine-tunes

only small sets of visual prompt embeddings, supplemented by a logit manipulation strategy to combat semantic drift and error propagation.

Lately, ViT-based architectures and backbones have been used to solve CI-SS trying to benefit from these architecture's efficiencies. On the one hand, two recent methods are based on a SwinB backbone (Liu et al. (2021)): first, CoinSeg (Zhang et al. (2023)) uses a discriminative feature representation thanks to inter-class and intra-class contrastive losses, while NeST (Xie et al. (2024)) proposes a pre-training method that transforms existing classifiers to initialize new ones, enhancing alignment with the model backbone. On the other hand, Incrementer (Shang et al. (2023)) takes advantage from the Segmenter (Strudel et al. (2021)) class embeddings to add new knowledge while retaining old information, based on a ViT-B backbone (Dosovitskiy et al. (2021)). It also proposes to adapt the $L_{KD}$ to only focus on old class regions and to regularise training alleviating both overfitting on new classes and confusion of similar semantic concepts. Since these methods are based on different backbones, they re-implement some previous methods using the same backbone for performance comparison such as MiB(ViT-B), RBC(ViT-B) or MicroSeg(SwinB).

### 2.3 Positioning of our method

At each step, the new data has most likely different statistical properties from the data seen previously. This *distributional shift* has a huge impact on the optimization of the model weights. This is enhanced in the case of CNNs where Batch Normalization (BN) layers are usually present, having a high dependency to the data distribution. Thus, if the distributional shift is not detected, the model will inevitably suffer from stronger catastrophic forgetting (Zhao et al. (2021)). Therefore, transformer-based architectures could be more appropriate for incremental learning. Indeed, Li et al. (2022) claim that they are better continual learners because they do not rely on BN.

Moreover, a very important aspect of real world applications is the memory footprint. Indeed, designing lightweight AI models is essential in edge computing environments, where optimizing resource usage for tasks like SS directly translates to more available computational power for other concurrent tasks, while preserving battery life. For instance, this is important for visual tasks on mobile phones, embedded systems in drones or autonomous robots relying on compact hardware. However, contrary to the non-incremental SS approaches where the efficiency and the performance drop using smaller backbones have been extensively studied, only big backbones (ViT-B and SwinB) have been tested for ViT-based CI-SS methods. Uncertainty on how the incremental strategies would perform with smaller backbones makes them impractical for real-world use. Besides, it would be interesting to build new strategies for these constrained cases.

Furthermore, during the training of previous SOTA approaches, the rigidity-elasticity trade-off is usually handled by a hyperparameter balancing the $L_{CE}$ of the predictions with relation to the new ground truth labels, and the $L_{KD}$ between the new and the old model's predictions. However, the values of the two losses vary depending on the confidence acquired by the old model, the number of classes for each task, and the number of images used for the training of each step. Previous models did not take this into account and applied a fixed hyperparameter independent of the task. In this work, we aim to tackle this, for a better trade-off between old and new classes, by introducing an adaptive balancing parameter during training, alongside a weighting branch merging parameter during inference.

To sum up our positioning, the proposed method TILES aims i) to present a solution for scenarios with severe memory constraints by exploiting the efficient transformer architectures for CI-SS; ii) to leverage the effectiveness of both knowledge distillation and expansion methods for incremental learning to tackle the semantic shift of the background iii) while minimizing the parameter expansion at each increment and iv) without retaining data for privacy concerns.

## 3 Method

### 3.1 Problem definition

The goal of CI-SS is to learn a model able to perform well on an increasing set of semantic classes. Let $T$ denote the total number of steps or increments at which a model has to learn a new task to cope with a new

subset of classes in addition to previously learned classes. At step $t \in \{1, \ldots, T\}$, let $D^t$ denote an additional subset of annotated data: $D^t = \{(x_i^t, y_i^t)\}_i$ where $x_i^t$ is the $i$-th image and $y_i^t$ is the corresponding ground truth of the same size, where each pixel is classified in $C^t$, the set of semantic classes at step $t$. The challenge is that the foreground classes are supervised only at one step, i.e. $C^n \bigcap C^m = \varnothing, \quad \forall n, m \in \{1, \ldots, T\}, n \neq m$, even if older classes continue to appear in new images or if future classes appeared in the older steps they are both considered as background. However, at the end, the model should still be able to retain knowledge on all seen classes $C^1 \bigcup C^2 \bigcup \ldots \bigcup C^T$.

## 3.2 Overview

We propose TILES as an expanding CI-SS approach that is convenient for scenarios with severe memory constraints. In fact, we prove that while it is possible for big encoders/decoders to encompass several semantic concepts learnt during different steps with minimum forgetting thanks to their considerable number of parameters, we argue that this is not possible when much smaller backbones are used, which we demonstrate in sec. 4.2. Therefore, we design TILES allowing limited expansion, particularly useful and necessary for these cases, while assuring a limited memory footprint. Similar to Incrementer (Shang et al. (2023)) our approach is based on the Strudel et al. (2021) architecture: Segmenter which uses a fully transformer encoder-decoder to generate class masks for SS. To adapt this architecture to CI learning, at each step $t$, we add $K_t$ class embeddings to represent the $K_t$ newly added classes $C^t$. This technique offers an efficient encoding of the class knowledge which could be saved in the next steps helping the model to retain information through its weights. We use a shared encoder between all tasks. However, we adopt an expanding architecture where each task has its specific specialized decoder branch $b$. The training of different steps of TILES is presented in Figure 2 (left). Note that the decoders are adapted so that the expansion is not prodigal in terms of number of parameters.

During inference, the input image is processed by the encoder, then, by each of the $T$ branches of the decoder in parallel. A final branch merging module is used to aggregate the predictions of different branches such as illustrated in Figure 3. Thanks to this expanding strategy, there is no semantic shift in each branch individually. The branch merging module carries out the choice of the final result using branch weights.

The different learning strategies in the initial step and then, how new branches are built on top of the previous model to segment all classes are detailed hereafter.

## 3.3 Encoder

The input image is denoted by $x_i^t \in \mathbb{R}^{H \times W \times C}$ where $H$, $W$ and $C$ are respectively the height, width and number of channels of the image. $x_i^t$ is divided into patches of size $P \times P$ pixels to generate the sequence of patches $\mathbf{x}_i^t = [\mathbf{x}_{i,1}^t, \ldots, \mathbf{x}_{i,N}^t] \in \mathbb{R}^{N \times P^2 \times C}$, where $N$ is the number of patches i.e. $N = \frac{HW}{P^2}$. These patches are then flattened, linearly projected and added to learnable position embeddings to generate the sequence $w_i^t = [w_{i,1}^t, \ldots, w_{i,N}^t] \in \mathbb{R}^{N \times D}$, $D$ being the embedding dimension. The encoder generates an output $z_i^t = [z_{i,1}^t, \ldots, z_{i,N}^t] \in \mathbb{R}^{N \times D}$. The encoder weights are updated at each increment.

## 3.4 Decoder incremental branches

To retain knowledge on classes, we use the learnable class embeddings as introduced in Segmenter (Strudel et al. (2021)) which are noted for the $b$-th branch (i.e. task) as $cls_b = [cls_b^1, \ldots, cls_b^{K_b}] \in \mathbb{R}^{K_b \times D}$. At the entry of each branch, the class embeddings $cls_b$ are concatenated to the encoder outputs $z_i^t$. Hence, the sequence fed into the Mask Decoder is $d_{i,b}^t = [z_{i,1}^t, \ldots, z_{i,N}^t, cls_b^1, \ldots, cls_b^{K_b}] \in \mathbb{R}^{(N+K_b) \times D}$. The self-attention block decoder (SAB) then generates the sequence $d_{i,b}'^t = [z_{i,1}'^t, \ldots, z_{i,N}'^t, cls_b'^1, \ldots, cls_b'^{K_b}] \in \mathbb{R}^{(N+K_b) \times D}$ made of the transformed class embeddings $cls_b' \in \mathbb{R}^{K_b \times D}$ and transformed encoded patches $z_i'^t \in \mathbb{R}^{N \times D}$. The class embeddings and encoded patches are then respectively linearly projected, and the $K_b$ masks are generated from their scalar product. For each branch, this output of dimension $N \times K_b$ is then upsampled to obtain a segmented image of the same size as the input in order to get the prediction at step $t$: $pred_{i,b}^t \in \mathbb{R}^{K_b \times H \times W}$ where $b \in \{1, \ldots, t\}$. As a result, each decoder branch is specialized in a task where classes from other tasks

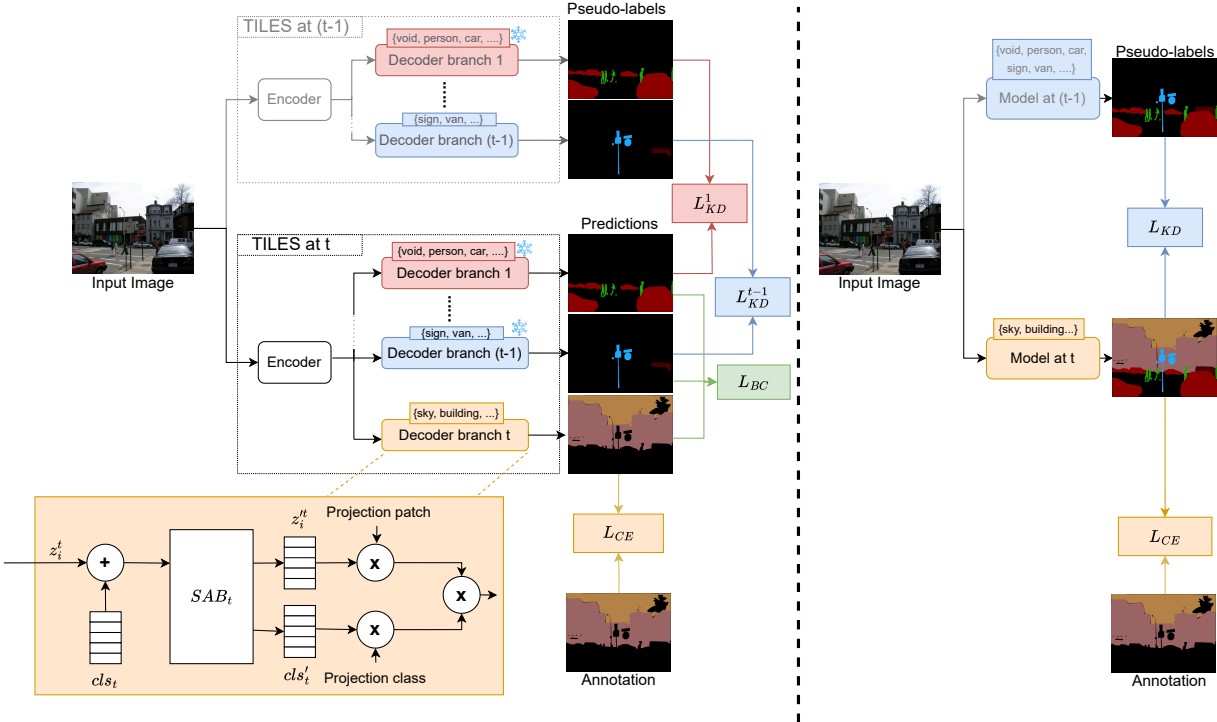

Figure 2: Training process of TILES (left) vs. KD-based SOTA methods (right). In TILES, the input image is first processed by the encoder, *e.g* ViT-Tiny, then each of the task branches (the tiles) generates a prediction map specific to the task it learned. $L_{KD}$ losses are computed between old and new predictions of each old branch separately to retain old knowledge, while $L_{CE}$ is used to learn to predict new classes via the new branch. $L_{BC}$ helps the differentiation and specification of each branch. In previous KD-based CI-SS approaches, both $L_{KD}$ and $L_{CE}$ are applied on the same output with possibly opposed goals. The architectural design is explained in sections 3.3, 3.4 and the learning strategy is detailed in section 3.6.

are considered as background. At step $t$, only the decoder weights of branch $b_t$ are changed while we freeze old step branches.

## 3.5   Branch merging

During inference, the predictions made by each task branch $pred_{i,b}^t$ are combined to generate the segmented image on all classes $pred_i^t$ such as presented in Figure 3. In particular, for each pixel, we merge the predictions $p^b$ of the different branches to decide the final value $p$ of that pixel using the following rules:

- if all task branches predict the pixel as background, i.e. $\{p^b = 0, \ \forall b \in \{1, \dots, T\}\}$, then the final label is the background: $p \leftarrow 0$ (label for background);

- if only one branch predicts the pixel as a foreground class, i.e. $\exists! \ b_1 \in \{1, \dots, T\}$ such that $\{p^{b_1} = c, \ c \neq 0\} \cap \{p^b = 0, \ \forall b \neq b_1\}$, then the final label is that foreground class: $p \leftarrow c$;

- if two or more branches predict the pixel as different foreground classes i.e. $\exists \ b_1, b_2 \in \{1, \dots, T\}, \ b_1 \neq b_2$ such that $\{p^{b_1} = c1, \ c1 \neq 0\} \cap \{p^{b_2} = c2, \ c2 \neq 0\}$ (red area in Figure 3), then the final label is the one scoring the highest confidence which is computed as a weighted probability $P(p^b) \cdot \gamma_b$, where $\gamma_b$ is the probability compensation weight for the branch $b$.

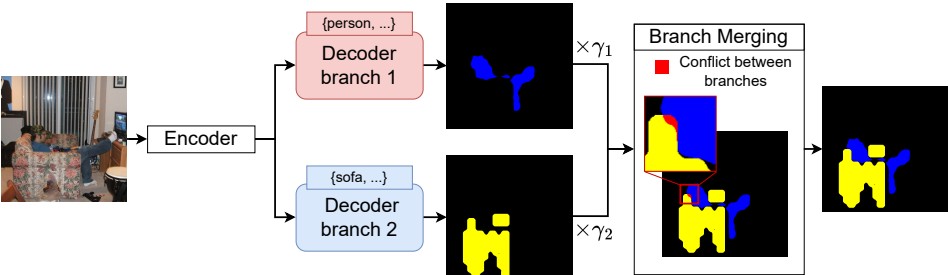

Figure 3: Branch merging module used during inference to merge two segmentation maps learned during different steps. Illustration is on a PascalVoc image using TILES-S trained on the [15-1] *overlapped* scenario. See sec. 3.5 for explanations.

**Probability compensation weight:** Background preponderance in images varies a lot depending on the dataset *e.g* no background class in ADE20k and a very present background class in Pascal-VOC such as detailed in sec. 4.1.1. It also highly depends on the nature of the task. For instance, the majority of pixels are considered as background in learning step 6 of the $100 - 10$ scenario, whereas background is much less present in the first step. This preponderance underrates, in different ratios, the probabilities of foreground classes, which causes different probability values of foreground classes while branch merging. To circumvent this, our branch merging strategy takes into account the relative sparsity of task classes $C^b$ by choosing $\gamma_b = \frac{\#p(background)}{\#p(images)}$, where $\#p(images)$ and $\#p(background)$ are respectively the number of pixels of all images used in train, and the corresponding number of background pixels.

### 3.6 Learning

Figure 2 illustrates the fundamental difference of loss computation between TILES and previous KD-based approaches. At a given step $t$, TILES (*left*) has $t$ branches specialized in predicting $t$ different tasks. It is trained to learn a new task on the new branch and to retain old tasks knowledge on old branches, while differentiating between semantic concepts of different tasks.

**Learn a new task:** To learn the new task, the ground-truth $y_i^t$ contains only labels belonging to the new set $C^t$ as foreground while all other classes are set to background. Hence, $L_{CE}(pred_{i,b=t}^t, y_i^t)$ is applied on the branch $t$ specific to the new task that estimates $pred_{i,b=t}^t$. The goal is to be able to distinguish between the new classes and the old ones which are considered as background in this branch.

**Retain knowledge on old tasks:** Since the encoder weights are shared between all tasks and are updated while learning the new task, we have to make sure that these weights will not change so much on the new background pixels (containing old classes' pixels) so that they will no longer fit the old tasks, leading to catastrophic forgetting. Therefore, $L_{KD}^b$ is computed for each branch $b \in \{1, .., t-1\}$ as a cross-entropy on the output softmax probabilities of predictions $pred_{i,b}^t$ and $pred_{i,b}^{t-1}$ at current and previous steps only on new background pixels (i.e. where $y_i^t = 0$). Such as introduced in Incrementer (Shang et al. (2023)), a $L_{KD}$ applied only on new background pixels instead of the whole image enables more elasticity on new foreground pixels ($y_i^t \neq 0$) to learn the new tasks and more rigidity on new background pixels which contain old foreground classes (see ablation in sec. 4.3).

**Differentiate between semantic concepts of different branches:** Incrementer (Shang et al. (2023)) demonstrated that close semantic concepts seen in different tasks could be confused since they are learnt in independent steps. This is further heightened in our case since we use totally independent decoders to learn different tasks. To alleviate this problem we add a Branch Classification Loss $L_{BC}$ defined as the mean values of the mask $M_i = \{m_{j,k}, 1 \leq j \leq W, 1 \leq k \leq H\}$ for each image $x_i \in \mathbb{R}^{H \times W \times C}$, where:

$$m_{j,k} = \begin{cases} 0 \; if \; p_{j,k}^b = 0 \; \forall \; b \; \in \{1,\dots,t\} \\ 0 \; if \; \exists! \; b_1 \in \{1,\dots,t\}; \{p_{j,k}^{b_1} = c, \; c \neq 0\} \cap \{p_{j,k}^b = 0, \; \forall b \neq b_1\} \\ 1 \; otherwise \end{cases} \tag{1}$$

$$L_{BC} = \frac{1}{WH} \sum_{j=1}^{W} \sum_{k=1}^{H} m_{j,k} \tag{2}$$

This loss encourages having at most one non-zero (foreground) value per pixel, alleviating confusion of close semantic classes learnt at different steps (see ablation in sec. 4.3).

**Total loss:** For step 1 the total loss is the presented $L_{CE}(pred_{i,1}^1, y_i^1)$. For steps $t > 1$, the total loss $Loss_i^t$ is computed as follows, where $\lambda_{old} > 0$ to balance rigidity vs. elasticity of the model:

$$Loss_i^t = \lambda_{old} \underbrace{\sum_{b \in [1:t-1]} L_{KD}^t(pred_{i,b}^t[y_i^t = 0], pred_{i,b}^{t-1}[y_i^t = 0])}_{\text{Loss on older tasks}} + \underbrace{L_{CE}(pred_{i,t}^t, y_i^t)}_{\text{Loss on the new task}} + \underbrace{L_{BC}(pred_{i,b}^1, .., pred_{i,b}^{t-1}, pred_{i,b}^t)}_{\text{Branch classification loss}}$$

$$\tag{3}$$

**Balancing losses:** At the beginning of each training step $t > 1$, we initialize the model parameters using those of the model at step $t - 1$. Hence, the different $L_{KD}$ losses are much smaller than the new $L_{CE}$. This causes the model to learn considerably the new task and forget the old one i.e. catastrophic forgetting, since the $L_{KD}$ have minor impact in the final loss. To balance the rigidity vs. the elasticity of the model, we set $\lambda_{old}$ to equalize the two terms at the beginning of training of each step: $\lambda_{old} = \frac{L_{CE}(iteration=1)}{L_{KD}(iteration=1)}$. This parameter takes into consideration the old model confidence on the old tasks and the nature of the tasks (number of images, preponderance of the background) which cause losses variations. Differently, previous KD-based methods (*right* in Figure 2) compute both $L_{KD}$ and $L_{CE}$ on the same output predicting all classes seen in tasks $\{1, .., t\}$ and balance them with a fixed weight per step which is a hyperparameter to find.

## 4 Experiments and results

### 4.1 Experimental settings

#### 4.1.1 Datasets, protocols and scenarios

To evaluate the effectiveness of TILES, we use two commonly used datasets, Pascal-VOC and ADE20k, and follow the evaluation protocols and scenarios associated.

**Pascal-VOC (Everingham et al. (2010)):** contains 11,530 images segmented and labeled into 20 possible semantic classes plus a background class which is highly predominant. Indeed, the segmentation is object-centric and 56% of all pixels are labeled as background. The results presented by previous methods differ as some approaches include the *background* pixels in the *mIoU* computation where others do not. We chose not to include it in the main table and we report the results of TILES while considering this value in the appendix (see Tables 2 and 4). Following previous work by Yang et al. (2022), we evaluate our models using both *disjoint* and *overlapped* protocols. In the *disjoint* protocol, each learning step contains a unique set of images, whose pixels belong to classes seen either in the current or previous step. Differently, in the *overlapped* protocol, each training step contains all the images that have at least one pixel of a novel class. Thus, images may contain pixels of classes that will be learned in the future. Notice that, since the sets of images are unique to each step in the *disjoint* protocol, the number of images used for training each step is drastically lower than in the *overlapped* protocol. The following three scenarios are evaluated for both protocols: $[19 - 1]$, $[15 - 5]$ and $[15 - 1]$ referring to [number of classes seen at step 1 - number of classes new seen at each step $> 1$ until reaching the 20 classes of Pascal-VOC].

**ADE20k (Zhou et al. (2019)):** is made of 20,000 images, each segmented into 150 possible classes. The segmentation is exhaustive in this case: images are segmented into stuff classes (*e.g* , *wall, sky*) and

things (*e.g* , *cars, person*). The *0-labeled* class does not carry semantic meaning, it only represents no distinguishable pixels that should be omitted in the training process (see Zhou et al. (2017)). TILES was trained to detect only the 150 classes (1-150). No clear details about how other methods consider this class. Like previous works (Shang et al. (2023); Zhang et al. (2023)), models are evaluated on ADE20k with the *overlapped* protocol since it is the more realistic one, on three scenarios: $[100-50]$, $[100-10]$ and $[50-50]$.

### 4.1.2   Implementation details

Similar to previous incremental approaches (Cermelli et al. (2020); Douillard et al. (2021)), random crops of size $512 \times 512$ pixels are used for training for both datasets. Moreover, the incremental learning configurations concerning the semantic classes learned at each step are retrieved from Cermelli et al. (2020). We optimize the models using SGD with a constant learning rate of 1e-3 throughout the steps. The models are trained with a batch size of 8, for 30 epochs for Pascal-VOC and for 64 epochs for ADE20k. The confidence weight $\gamma_b$ and the loss weight on old tasks $\lambda_{old}$ are computed online (*c.f* as explained in sections 3.5 and 3.6).

We compute three variants of TILES with suffixes -B, -S and -T denoting Base, Small or Tiny. In fact, we use a ViT (Dosovitskiy et al. (2021)) pre-trained on Imagenet (Deng et al. (2009)) for image classification (-B, -S or -T) as encoder. TILES-T uses the same encoder and decoder as Segmenter-T (Strudel et al. (2021)), resulting in 6.7M (million) parameters for one step and adding 0.4M parameters at each subsequent step as we add a branch decoder for each increment. For scalability reasons, TILES-S and TILES-B use the original ViT-B and ViT-S encoders respectively but adopt a custom decoder having a smaller hidden size of 256. This reduction helps limit the expansion of parameters for many step's scenarios. A dense layer is used to resize the resp. 384 and 768 hidden sizes output of ViT-S and ViT-B encoders to 256. Thus, TILES-S (TILES-B) uses 23.8M (88.2M) parameters at the initial step with additional 1.8M parameters at each step for both configurations. Table 5 details the number of parameters used for each scenario for TILES and previous methods.

For Incrementer (Shang et al. (2023)), we use the same values reported in the their paper, where ViT-B encoder and decoders are used. We perform the missing experiments for the ViT-S and ViT-T variants of Incrementer for comparison with TILES variants using the same encoder backbones.

### 4.1.3   Evaluation metrics

The evaluation metric usually used for CI-SS is the *mean Intersection over Union (mIoU)* which is the mean of the IoU per class. Results of the last step models are compared: a mIoU score is computed for classes learned in the first step, another for those learnt in subsequent step(s), and the *all* column (in the tables hereafter) represents the average mIoU over all classes.

**Knowledge Remaining (KR):** Different approaches use different models and backbones, which are based on different *joint* performances (see Table 4): the non-incremental scenario in which the model learns from the whole data at once which can be considered as an upper bound for a given architecture and backbone. Hence, it is not fair to evaluate absolute *all mIoU* performances to evaluate the gain of the incremental strategies, while using different architectures. To this end, we propose an additional evaluation metric: the *Knowledge Remaining (KR)*. This metric is actually inspired by work done by Hoyer et al. (2022) in domain adaptation to fairly compare adaptation strategies while using different architectures and backbones. It is calculated as the ratio of the performance in the incremental setup by the performance in the *joint* setup for each scenario: $KR = \frac{all}{mIoU(Joint)}$.

### 4.2   Results

We compare in Table 1, Table 2 and Table 3 different state-of-the-art CI-SS methods on the predefined scenarios and protocols of Pascal-VOC and ADE20k datasets. Since these methods are based on different architectures and backbones, their corresponding *joint* setups have different values such as illustrated in Table 4. Moreover, it is important to highlight that the memory footprint varies a lot depending on the backbone, which has not been considered by previous CI-SS methods, by comparing absolute *mIoU* while having different *joint* values and model sizes. For these reasons and to ensure a fair comparison between the

| Backbone | Method | #P (M) | 19-1 (2 steps) | | | | 15-5 (2 steps) | | | | 15-1 (6 steps) | | | |
|---|---|---|---|---|---|---|---|---|---|---|---|---|---|---|
| | | | 1-19 | 20 | all | KR | 1-15 | 16-20 | all | KR | 1-15 | 16-20 | all | KR |
| ResNet-101 | MicroSeg[+] | 66 | 78.8 | 14.0 | 75.7 | 97.4 | 80.4 | 52.8 | 73.8 | **95.0** | 80.1 | 36.8 | 69.8 | 89.8 |
| | Bg_Adapt[+] | 66 | 78.2 | 42.2 | 76.4 | **98.7** | - | - | - | - | 77.6 | 45.9 | 79.4 | **90.4** |
| ViT-B | Incrementer | 102 | 82.5 | 61.0 | 81.4 | **99.4** | 82.5 | 69.3 | 79.2 | 96.7 | 79.6 | 59.7 | 74.6 | 91.1 |
| | TILES-B | 90 to 97 | 81.0 | 53.4 | 79.6 | **99.4** | 81.9 | 69.4 | 78.8 | **98.4** | 79.8 | 55.4 | 73.7 | **92.0** |
| ViT-S | Incrementer | 26 | 76.4 | 47.5 | 75.0 | 95.4 | 74.0 | 56.1 | 69.5 | 88.4 | 72.6 | 46.1 | 66.0 | 84.0 |
| | TILES-S | 26 to 33 | 79.1 | 53.3 | 77.8 | **99.0** | 79.7 | 65.3 | 76.1 | **96.8** | 77.1 | 47.8 | 69.8 | **88.8** |
| ViT-T | Incrementer | 7 | 67.2 | 39.7 | 65.8 | 90.4 | 56.7 | 38.0 | 52.0 | 71.4 | 53.3 | 20.0 | 45.0 | 61.8 |
| | TILES-T | 7 to 9 | 73.6 | 38.0 | 71.8 | **98.6** | 71.7 | 47.2 | 65.6 | **90.1** | 66.3 | 21.1 | 55.0 | **75.5** |

Table 1: CI performances (mIoU and KR in %) on Pascal-VOC for different *overlapped* scenarios. Best KR per backbone per scenario in **bold**. [+] mIoU computation is biased by considering background IoU. **#P** is the number of parameters (in millions). A more complete comparison is in the appendix (Table 1).

| Backbone | Method | #P (M) | 19-1 (2 steps) | | | | 15-5 (2 steps) | | | | 15-1 (6 steps) | | | |
|---|---|---|---|---|---|---|---|---|---|---|---|---|---|---|
| | | | 1-19 | 20 | all | KR | 1-15 | 16-20 | all | KR | 1-15 | 16-20 | all | KR |
| ResNet-101 | RBC | 66 | 76.4 | 45.8 | 74.9 | **96.4** | 75.1 | 49.7 | 68.8 | **88.5** | 61.7 | 19.5 | 51.1 | 65.8 |
| | RCIL[+] | 66 | - | - | - | - | 75.0 | 42.8 | 67.3 | 87.0 | 66.1 | 18.2 | 54.7 | **70.7** |
| ViT-B | Incrementer | 102 | 82.4 | 64.2 | 81.5 | **99.5** | 81.6 | 62.2 | 76.8 | **93.8** | 81.4 | 57.1 | 75.3 | **91.9** |
| | TILES-B | 90 to 97 | 80.5 | 55.2 | 79.2 | 98.9 | 77.6 | 49.3 | 70.5 | 88.0 | 74.1 | 35.7 | 64.5 | 80.5 |
| ViT-S | Incrementer | 26 | 76.4 | 19.9 | 73.6 | 93.6 | 75.2 | 26.9 | 63.1 | 80.3 | 71.9 | 38.6 | 63.6 | 80.9 |
| | TILES-S | 26 to 33 | 79.1 | 51.5 | 77.7 | **98.9** | 76.1 | 47.7 | 69.0 | **87.8** | 73.9 | 35.1 | 64.2 | **81.7** |
| ViT-T | Incrementer | 7 | 68.9 | 13.6 | 66.1 | 90.8 | 65.7 | 22.9 | 55.0 | 75.5 | 59.1 | 14.2 | 47.9 | 65.8 |
| | TILES-T | 7 to 9 | 72.9 | 44.5 | 71.5 | **98.2** | 68.5 | 37.4 | 60.7 | **83.4** | 61.6 | 26.5 | 51.7 | **71.0** |

Table 2: CI performances (mIoU and KR in %) on Pascal-VOC for different *disjoint* scenarios. Best KR per backbone per scenario in **bold**. [+] mIoU computation is biased by considering background IoU. **#P** is the number of parameters (in millions). A more complete comparison is in the appendix (Table 3).

| Backbone | Method | #P (M) | 100-50 (2 steps) | | | | 100-10 (6 steps) | | | | 50-50 (3 steps) | | | |
|---|---|---|---|---|---|---|---|---|---|---|---|---|---|---|
| | | | 1-100 | 101-150 | all | KR | 1-100 | 101-150 | all | KR | 1-50 | 51-150 | all | KR |
| ResNet-101 | UCD | 66 | 40.4 | 27.3 | 36.0 | **92.5** | 28.6 | 12.4 | 23.2 | 59.6 | 39.3 | 22.2 | 27.9 | 71.7 |
| | RBC | 66 | 42.9 | 21.5 | 35.8 | 92.0 | 39.0 | 21.7 | 33.2 | 85.3 | 49.6 | 26.3 | 34.1 | **87.7** |
| | Bg_Adapt[+] | 66 | 42.0 | 23.0 | 35.7 | 91.8 | 41.1 | 23.1 | 35.2 | **90.5** | 47.9 | 26.5 | 33.7 | 86.6 |
| ViT-B | Incrementer | 102 | 49.4 | 35.6 | 44.8 | 93.1 | 48.5 | 34.6 | 43.6 | **91.3** | 56.2 | 37.8 | 43.9 | 91.3 |
| | TILES-B | 90 to 97 | 51.9 | 39.3 | 47.7 | **99.2** | 50.6 | 18.6 | 39.9 | 82.7 | 58.9 | 41.2 | 47.1 | **97.9** |
| ViT-S | Incrementer | 26 | 48.7 | 29.9 | 42.4 | 90.6 | 42.3 | 15.1 | 33.2 | 70.9 | 55.7 | 32.6 | 40.3 | 86.1 |
| | TILES-S | 26 to 33 | 50.2 | 34.1 | 44.8 | **98.2** | 46.7 | 15.8 | 36.4 | **79.8** | 55.5 | 37.4 | 43.4 | **95.2** |
| ViT-T | Incrementer | 7 | 47.6 | 8.6 | 34.6 | 89.9 | 34.1 | 8.6 | 24.6 | 63.9 | 49.9 | 23.4 | 32.2 | 83.6 |
| | TILES-T | 7 to 9 | 43.1 | 24.8 | 37.0 | **96.1** | 36.8 | 16.8 | 30.1 | **78.2** | 50.1 | 28.1 | 35.4 | **91.9** |

Table 3: CI performances (mIoU and KR in %) on ADE20k for different *overlapped* scenarios. Best KR per backbone per scenario in **bold**. [+] mIoU computation is biased by considering background IoU. **#P** is the number of parameters (in millions). A more complete comparison is in the appendix (Table 5).

different incremental techniques, we focus on comparing the *knowledge remaining* (KR) which evaluates the capacity of retaining old knowledge while learning new tasks, regardless of the used architecture. The goal here is to fairly compare the different approaches, and to propose a new light-weight option for use-cases with severe memory constraints. Note that methods with [+] consider the background in their *mIoU* computation which distorts results since generally all methods have very good background *IoU*.

On the one hand, we can notice in the three tables and their complete versions in the appendix that previous ViT based approaches assure in general better absolute and KR performances than the CNN based approaches. In addition to the incremental techniques used, this is also due to the fact that visual transformers are better continual learners than CNNs such as discussed in sec. 2.3, to the better *joint* values and to the bigger memory footprint used. In fact, using more parameters improves the models' capacity to encompass the old and the new knowledge with minimum forgetting. Nevertheless, despite using less parameters than previous ViT based approaches thanks to the smaller decoder (see sec. 4.1.2 for details), TILES-B is able to achieve interesting results both datasets.

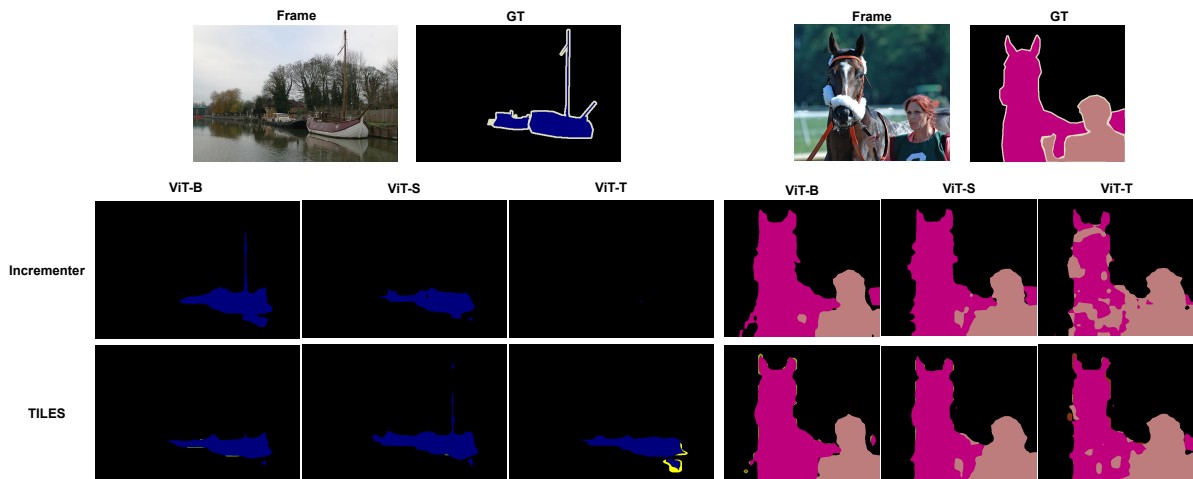

Figure 4: Samples of frames, their Ground Truth (GT) annotations, and corresponding predictions using Incrementer and TILES with 3 different backbones each for the Pascal-VOC 15-1 *overlapped* scenario.

On the other hand, we compare Incrementer (Shang et al. (2023)) with TILES using smaller backbones to study behaviors when severe memory constraints are applied. We choose Incrementer as a reference because it has best results among most scenarios on both ADE20k and Pascal-VOC datasets, and because it uses the same semantic segmentation base method as TILES: Segmenter (Strudel et al. (2021)). We can see that fine-tuning parameters when using smaller backbones results in a big performance drop since the limited number of parameter is not able to encompass both old and new knowledge while ensuring a good rigidity vs. elasticity trade-off. As an alternative, TILES-S and TILES-T provide always better performances, with a KR percentage points (p.p.) difference ranging from: i) for ViT-S from 0.8 to 8.4 for Pascal-VOC and from 7.6 to 9.1 for ADE20k, and ii) for ViT-T from 5.2 to 18.4 for Pascal-VOC and from 6.2 to 14.3 for ADE20k. Indeed, the performance gap between TILES and Incrementer is heightened for smaller backbones. These improvements are especially thanks to the adopted expanding mechanism which seems necessary to learn new tasks without big forgetting, while adding a limited number of parameters at each step. Indeed, depending on the balance between the old and new losses, applying smaller backbones to Incrementer seems to either cause catastrophic forgetting or to limit learning new tasks. This can be seen in Figure 4 that TILES maintains relatively strong performance with smaller backbones, whereas Incrementer exhibits a more significant performance drop, especially with the ViT-T backbone.

Figure 1 illustrates the performance of TILES and Incrementer according to the number of parameters of the used model for the Pascal-VOC 15-5 *overlapped* scenario. It demonstrates that TILES strongly outperforms Incrementer when considering models with comparable sizes. We also include error bars in this figure for TILES demonstrating the small variations through iterations. Similar figures are included in the appendix for the other scenarios and datasets (Figures 1, 2, and 3). Note that, despite adopting an expansion mechanism, the number of parameters added in all these cases is paramount (see Table 5 for details of number of parameters and the Table 6 in the appendix for further memory analysis). In fact, TILES-B uses less parameters than other ViT-B and SwinB based approaches thanks to the light-weight decoder used. For TILES-S and TILES-T, we can prove efficiency by providing major improvements compared to Incrementer while adding 1.8M and 0.4M parameters per step respectively.

It is also important to highlight that TILES-S achieves the same absolute results as Incrementer (ViT-B) for the $[100 - 50]$ and $[50 - 50]$ ADE20k protocols despite displaying different *joint* performances and while using up to 4 times fewer parameters (25 vs. 102) thanks to the adopted expanding mechanism along with the corresponding losses and branch merging module. Similarly, TILES-T achieves and even surpasses CNN based approaches in some cases, despite the smaller *joint* performances and while using up to 9 times fewer parameters (7.1 vs. 66). TILES-T also provides similar or better absolute performances compared to SwinB-based methods while using up to 14 times fewer parameters (7.1 vs. 104) for the same ADE20k protocols.

| Model | Backbone | Pascal-VOC | ADE20k |
|---|---|---|---|
| Deeplab-v3 | ResNet-101 | 77.4 | 38.9 |
| Mask2Former | ResNet-101 | - | 43.1 |
| Deeplab-v3 | SwinB | 82.7 | 39.1 |
| Incrementer | ViT-B | 81.9 | 48.1 |
| TILES-B | ViT-B | 80.1 | 48.1 |
| Incrementer | ViT-S | 78.6 | 46.8 |
| TILES-S | ViT-S | 78.6 | 45.6 |
| Incrementer | ViT-T | 72.8 | 38.5 |
| TILES-T | ViT-T | 72.8 | 38.5 |

Table 4: Performance (mIoU in %) of the *joint* setting of different models and backbones for Pascal-VOC and ADE20k datasets.

| Model | Backbone | 1 step | 2 steps | 3 steps | 6 steps |
|---|---|---|---|---|---|
| Deeplab-v3 | ResNet-101 | 66 | 66 | 66 | 66 |
| Mask2Former | ResNet-101 | 63 | 63 | 63 | 63 |
| Deeplab-v3 | SwinB | 104 | 104 | 104 | 104 |
| Incrementer | ViT-B | 102 | 102 | 102 | 102 |
| TILES-B | ViT-B | 88 | 90 | 92 | 97 |
| Incrementer | ViT-S | 26 | 26 | 26 | 26 |
| TILES-S | ViT-S | 24 | 26 | 27 | 33 |
| Incrementer | ViT-T | 6.7 | 6.7 | 6.7 | 6.7 |
| TILES-T | ViT-T | 6.7 | 7.1 | 7.5 | 8.7 |

Table 5: Number of parameters (in million) used with relation to the number of steps of the scenario.

This proves the over-allocation of parameters by previous CI-SS methods and the importance of studying the efficacy of models for highly constrained tasks.

Moreover, despite being an expanding method, TILES-T shows a good scalability with the number of increments (239 steps would be necessary to surpass the $102M$ parameters used by previous ViT-based approaches). Therefore, TILES-T is convenient for applications with extremely severe memory constraints or needing a large number of increments. Besides, TILES-S shows closest performances to previous ViT-based methods while keeping the number of parameters lower until an expansion of 43 increments. Thus, TILE-S is adapted to applications requiring a lower number of increments, and where improved performance is more important than severe memory constraints. In the absence of prior work using small backbones, the main goal of proposing TILES is to offer a first compelling alternative for scenarios with certain conditions. Indeed, the choice of architecture depends on the acceptable trade-off between performance and resource usage, as well as the maximum number of increments required by each application. However, beyond the 239 threshold, we acknowledge that we cannot assert whether TILES-T or Incrementer (ViT-B) is the more effective choice. While scenarios with increments exceeding these thresholds cause the augmentation of the memory footprint needed for TILES, we have no guarantee that Incrementer can be the better choice and whether it can handle catastrophic forgetting while updating both the encoder and the decoder parameters for hundreds or thousands of increments. We encourage the research community for a deeper investigation of these use cases.

### 4.3 Ablation study

| Method | #P (M) | 100-50 (2 steps) | | | | 100-10 (6 steps) | | | | 50-50 (3 steps) | | | | Joint |
|---|---|---|---|---|---|---|---|---|---|---|---|---|---|---|
| | | 1-100 | 101-150 | all | KR | 1-100 | 101-150 | all | KR | 1-50 | 51-150 | all | KR | |
| TILES-S (SS-D) | 30 to 46 | 49.8 | 35.2 | 44.9 | 95.9 | 42.1 | 31.6 | 38.6 | 82.5 | 56.1 | 41.2 | 43.6 | 93.2 | 46.8 |
| TILES-S (ours) | 26 to 33 | 50.2 | 34.1 | 44.8 | 98.2 | 46.7 | 15.8 | 36.4 | 79.8 | 55.5 | 37.4 | 43.4 | 95.2 | 45.6 |

Table 6: Influence of decoder architecture on TILES-S performance (mIoU and KR in %) on 3 *overlapped* scenarios on ADE20k. SS-D denotes Segmenter-S decoder. **#P** is the number of parameters (in millions).

**TILES-S using Segmenter-S decoder:** In Table 6, we compare ADE20k results using TILES-S but with two different decoders: Segmenter-S decoders which add 4M parameters at each step and our light-weight decoders adding $1.8M$ parameters per step. It demonstrates that, despite the much bigger Segmenter-S decoder, the absolute $mIoU$ improvement compared to the TILES-S decoder is null or minor compared to the memory footprint increase (ranging from 15% to 48%). This proves that the adopted decoder architecture is sufficient to encompass the new knowledge while adding a limited memory footprint at each step.

| Ablation | $\lambda_{old}$ | $L_{KD}$ | $\gamma_b$ | $L_{BC}$ | 1-15 | 16-20 | all |
|---|---|---|---|---|---|---|---|
| TILES | $\neq 1$ | on new background pixels | $\neq 1$ | ✓ | 71.7 | 47.2 | **65.6** |
| A1 | $= \mathbf{1}$ | on new background pixels | $\neq 1$ | ✓ | 67.2 | 55.0 | 64.2 |
| A2 | $\neq 1$ | *on all pixels* | $\neq 1$ | ✓ | 71.5 | 42.6 | 64.3 |
| A3 | $\neq 1$ | on new background pixels | $= \mathbf{1}$ | ✓ | 71.0 | 45.1 | 64.5 |
| A4 | $\neq 1$ | on new background pixels | $\neq 1$ | ✗ | 72.4 | 42.7 | 65.0 |

Table 7: Ablation study of loss balancing $\lambda_{old}$ (A1), applying $L_{KD}$ on all pixels or only on new background pixels (A2), probability compensation weight $\gamma_b$ (A3) and binary classification loss $L_{BC}$ (A4) on TILES-T performance (mIoU in %) on Pascal-VOC [15-5] *disjoint* scenario.

**Balancing losses:** Table 7 (A1 and TILES) shows that loss balancing in TILES ($\lambda_{old} \neq 1$ as detailed) is beneficial (+1.4 p.p. $mIoU$ for TILES-T on Pascal-VOC $[15-5]$ *disjoint*). We can notice that equalizing losses at the beginning, alleviates forgetting and creates a better rigidity vs. elasticity trade-off. This can be even more important for several-step scenarios where the model forgets old knowledge at each increment.

**Applying $L_{KD}$ on all pixels:** Table 7 (A2 and TILES) proves that applying the knowledge distillation loss $L_{KD}$ only on new background pixels improves significantly the new classes performances compared to applying this loss to whole image. In fact, this technique provides more elasticity on the pixels corresponding to new classes by retaining knowledge only on the new background pixels that could be potentially old classes learnt in old steps.

**Compensation weight of probability for branch merging:** Table 7 (A3 and TILES) shows that compensating the branch prediction probabilities is beneficial as it gives more weight to ignored branches i.e. in this case the new branch as the training-set for this step is smaller. Same conclusions can be found for the ADE20k 100-50 scenario with $-1.3$ mIoU p.p. for TILES-T and $-0.4$ p.p. for TILES-S.

**Impact of branch classification loss:** Since different decoders are used for different tasks for TILES, semantically close concepts could be learnt by separate decoders causing a confusion between them and thus a performance drop (see sec. 3.6). Table 7 (A4 and TILES) shows the big degradation of new classes performances if the branch classification loss between the branches $L_{BC}$ is retrieved. Moreover, removing this component from TILES-T for the ADE20k 100-50 scenario causes $-1.9$ mIoU p.p. drop.

## 5 Conclusion

In this work, we elaborated a complete comparison across previous SOTA CI-SS methods based on different backbones. While different performance trends can be remarked, all these methods use quite large memory footprint without any study about their efficiency with regards this aspect. Thus, we proposed TILES, a new CI learning method based on a ViT architecture for SS and specifically convenient for use cases with severe memory constraints. Indeed, we demonstrated a big performance drop for a previous SOTA method when smaller backbones are used, unlike TILES which is more adapted for these cases. Moreover, TILES can even outperform previous models which use much bigger backbones (up to $14\times$ bigger) when comparing absolute performance. Since TILES proposes the first lightweight method for CI-SS, we hope that this work stimulates the AI community's interest to study models efficiency for CI-SS.

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
