# OpenReview forum: "Efficient Class-Incremental Segmentation Learning via Expanding Visual Transformers"
_TMLR — Rejected by TMLR_

### Review · Reviewer_tUFz · 2025-06-07

**Summary Of Contributions:**

This paper addresses class-incremental semantic segmentation (CI-SS) with a focus on memory-constrained scenarios. The authors propose TILES (Transformer-based Incremental Learning for Expanding Segmenter), which uses an expanding architecture with task-specific decoder branches built on small Vision Transformer backbones. The main contributions include: (1) a fair comparison study of existing CI-SS methods across different backbone sizes, (2) demonstration that existing ViT-based methods perform poorly with smaller backbones, (3) introduction of TILES with expanding decoder branches, adaptive loss balancing, and branch merging strategies, and (4) a new evaluation metric called Knowledge Remaining (KR) to fairly compare methods with different architectures.

**Audience:**

Yes

**Broader Impact Concerns:**

While this paper does not include a Broader Impact Statement, there are no critical broader impact concerns since this paper is fundamentally about algorithmic efficiency rather than introducing new capabilities that could raise novel ethical concerns.

**Claims And Evidence:**

Yes

**Requested Changes:**

### Critical Changes:

- Add comprehensive memory analysis including actual memory usage, FLOPs, and computational overhead comparisons, not just parameter counts.
- Provide complete mathematical formulation for the branch classification loss L_BC and formal definition of the Knowledge Remaining (KR) metric.
- Include comparison with recent methods like CoMFormer and ECLIPSE to establish current state-of-the-art baselines.
- Clarify reproduction methodology for baseline methods, particularly hyperparameter tuning strategies when adapting to smaller backbones.

### Recommended Improvements:

- Deeper technical analysis explaining why TILES-B underperforms Incrementer-B and whether proposed components are specialized for lightweight encoders.
- More detailed ablation studies on probability compensation and branch classification loss components.
- Providing a qualitative comparison with existing methods.
- Computational overhead analysis during training and inference phases.
- Better motivation for the probability compensation mechanism beyond engineering tricks.
- Discuss on limitations of the proposed method.

**Strengths And Weaknesses:**

### Strengths:

- Addresses an important practical problem of memory-constrained incremental learning.
- Provides a systematic comparison across different backbone sizes, highlighting a gap in existing literature.
- Demonstrates significant improvements over existing methods when using small backbones (ViT-S, ViT-T).
- Comprehensive experimental evaluation on standard benchmarks (Pascal-VOC, ADE20k).
- Introduces expanding architecture that maintains specialized branches for different tasks.

### Weaknesses:

- Insufficient Memory Analysis: The paper repeatedly claims memory efficiency but only reports parameter counts. Critical metrics like actual memory usage, FLOPs, and computational overhead are missing, making it impossible to verify the claimed memory advantages.
- Limited Technical Novelty: The method heavily borrows from Incrementer with minimal original contributions. The core architecture (encoder-decoder, class embeddings incremental pipeline, KD regularization) is largely identical. The main differences, i.e., probability compensation and branch classification loss, are incremental improvements rather than fundamental innovations.
- Unclear Technical Details:
  - The branch classification loss L_BC lacks a proper mathematical formulation, without a detailed definition.
  - The mask definition in Equation 1 uses m_{j,k} but never explains how it connects to the actual loss computation.
  - Probability compensation appears over-engineered and dataset-dependent, lacking technical novelty.
- Questionable Evaluation Methodology:
  - The Knowledge Remaining (KR) metric lacks a formal definition or mathematical formulation.
- Inconsistent Performance Analysis:
  - TILES-B underperforms Incrementer-B across most scenarios, raising questions about whether the proposed components are truly beneficial or only work for lightweight encoders. The paper lacks an adequate explanation for this performance discrepancy.
  - This result also raises the reproduction concerns. It is unclear if hyperparameter tuning was performed when adapting Incrementer to smaller backbones.
- Missing Recent Comparisons: The paper omits comparison with recent transformer-based CI-SS methods like CoMFormer*(CVPR'23) and ECLIPSE* (CVPR'24), limiting the comprehensiveness of the evaluation.
  * Cermelli, Fabio, Matthieu Cord, and Arthur Douillard. "Comformer: Continual learning in semantic and panoptic segmentation." Proceedings of the IEEE/CVF Conference on Computer Vision and Pattern Recognition. 2023.
  * Kim, Beomyoung, Joonsang Yu, and Sung Ju Hwang. "Eclipse: Efficient continual learning in panoptic segmentation with visual prompt tuning." Proceedings of the IEEE/CVF Conference on Computer Vision and Pattern Recognition. 2024.

---

> ### Author Response · Authors · 2025-07-05
> **We thank the reviewer for its valuable comments. Changes in the main paper are highlighted in red.**
>
> - As requested, further memory analysis has been added to the Sec.2 of the appendix.
>
> - The goal of TILES is to propose an alternative to the current SOTA approach which is Incrementer, by demonstrating the limitations of parameter update when using small backbones and proving the importance of model expansion in these cases. For this reason, we use the same segmentation architecture to focus on comparing the gain brought by the expanding architecture and the branch merging technique rather than proposing a completely novel architecture.
>
> - For more clarity, we added the mathematical formulation (cf. Formula 2) of loss corresponding to the already present description: "$L_{BC}$ defined as the mean values of the mask $M_i=\{m_{j,k}, 1 \le j \le W, 1 \le k \le H \}$ for each image $x_i \in \mathbb{R}^{H \times W \times C}$".
>
> - Since different decoders are learnt during different steps, confidence values of different tasks can have different margins. To circumvent this, and without adding any hyper-parameter to tune during training different scenarios, we propose the probability compensation, to take into account the relative sparsity of task classes which differ a lot depending on the dataset and the scenario.
>
> - For more clarity, we added in Sec.4.1.3 the mathematical formulation corresponding to the definition of KR: "It is calculated as the ratio of the performance in the incremental setup by the performance in the joint setup."
>
> - For TILES-B, we use lightweight decoders such as explained in the implementation details. The number of parameters, FLOPs, and memory at inference are smaller than those for Incrementer for all the tested scenarios (see Tab. 5 in the main paper and Tab. 6 in the appendix). The joint performance for TILES-B is then also lower than Incrementer for Pascal-VOC (-1.8 mIoU p.p.). That explains better KR performance for TILES-B regardless of the lower all values. Experiments conducted using the same heavier decoder architecture as Incrementer for TILES-B demonstrated better results than Incrementer. However, we chose not to include these values as, in TILES, priority is given to sparing the number of parameters while maintaining comparable performance.
>
> - Hyper-parameter tuning was not performed neither for TILES nor Incrementer. In fact, for Incrementer, we follow the hyper-parameters specified in the main paper which seem to be effective for the different scenarios: "we train our method on Pascal VOC with learning rate 1e-4 for 32 epochs and ADE20k with 1e-3 for 64 epochs, and the learning rate is half of the initial value in the following steps" (Shang et al. (2023)). Other hyper-parameters were made hidden and seem manually tuned depending on the scenario as the pattern to define them is unclear: "For the single-class per step protocols, we reduce the learning rate and iterations in part of steps to prevent overfitting" (Shang et al. (2023)). For this reason, we keep the same values found for the ViT-B model. For TILES, we keep the same learning rate for all experiments, and other hyper-parameters are computed automatically ($\gamma_b$ and $\lambda_{old}$) making adaptation to new datasets and scenarios easier.
>
> - We thank the reviewer for the valuable references (ComFormer and ECLIPSE). We included their descriptions in the related work section and their performance in Tab. 5 of the appendix since they only provide CI-SS results for ADE20k. We separate these methods in a new block as they are based on a Mask2Former architecture using a ResNet101 backbone. New values have been added to Tables 4 and 5 in the paper to give the joint performance and the number of parameters for this setup.
>
> - In the main paper, we show ablation studies on the probability compensation and the branch classification loss components on the Pascal-VOC 15-5 disjoint scenario, showing their importance especially for learning new tasks. Other experiments on the ADE20k 100-50 scenario demonstrate the same conclusions and have been added to the main paper (see section 4.3).
>
> - We added Fig. 4, which presents segmentation outputs from both Incrementer and TILES on two samples for the Pascal-VOC 15-1 overlap scenario. The figure illustrates that TILES maintains relatively strong performance with smaller backbones, whereas Incrementer exhibits a more significant performance drop, especially with the ViT-T backbone.
>
> - Further discussion on the limitations of the proposed method (especially the memory expansion) have been added to the paper (see last paragraph in Sec. 4.2).

---

> > ### Comment · Reviewer_tUFz · 2025-07-21
> > **One more Clarification**
> >
> > I appreciate the authors for providing comprehensive and detailed rebuttals.
> >
> > I would like to request one additional clarification regarding the decoder architecture. My understanding is that Incrementer increases the number of class tokens at each incremental step while maintaining the same encoder and decoder architectures in terms of model size. In contrast, TILES incrementally expands the decoder itself—thus increasing its model size—while keeping the encoder architecture unchanged. As a result, the final TILES model requires more parameters and incurs higher GFLOPs.
> >
> > Could the authors please confirm whether this interpretation is correct?

---

> > > ### Author Response · Authors · 2025-07-21
> > > **Response for claification**
> > >
> > > We do confirm that this interpretation is correct.
> > > Both Incrementer and TILES are based on Knowledge Distillation but only TILES is using architecture expansion technique on the decoder part.
> > > While Incrementer finetunes all parameters (from encoder and decoder) at every step, TILES finetunes only parameters from encoder and the new decoder.
> > > Paragraph 2 of Section 4.1.2 details the increase of parameters per step, depending on the backbone used: +1.8M parameters per step for TILES-B and TILES-S; +0.4M parameters for TILES-T.
> > > Table 5 also summarizes the number of parameters of both models (Incrementer and TILES) depending on the number of steps.
> > >
> > > As shown in Tables 1,2,3, the no-expanding strategy of Incrementer on a model with lots of parameters (i.e., using ViT-B backbone) can handle the learning of new knowledge.
> > > However, in case of severe memory constraints, we demonstrate through the different results that this expansion strategy is more adapted.
> > > TILES-S and TILES-T are more able to encompass both old and new knowledge (i.e., ensuring a good rigidity vs. elasticity trade-off) than their counterpart (ViT-S) and (ViT-T) Incrementer models:
> > >
> > > - For ViT-S: a gain from +0.8 to +8.4 p.p. in KR for Pascal-VOC and from +7.6 to +9.1 p.p. in KR for ADE20k
> > > - For ViT-T: a gain from +5.2 to +18.7 p.p. in KR for Pascal-VOC and from +6.2 to +14.3 p.p. in KR for ADE20k.
> > >
> > > Figure 1 also illustrates (for the Pascal-VOC 15-5 overlapped scenario) the performance gain of TILES compared to Incrementer, while having always a comparable number of parameters.

---

### Review · Reviewer_rhM8 · 2025-06-19

**Summary Of Contributions:**

This paper introduces TILES (Transformer-based Incremental Learning for Expanding Segmenter) for class-incremental segmentation. Building upon Segmenter, the authors propose a network expansion method. Specifically, when a new task is given, a new decoder is assigned to learn class-specific information. To prevent forgetting and background semantic shift, the decoders from previous tasks are also reused to output pseudo labels, which are then used to update the encoder through KD loss. Extensive experiments on various datasets demonstrate the effectiveness of the proposed method.

**Audience:**

Yes

**Claims And Evidence:**

Yes

**Requested Changes:**

Please refer to the main weaknesses.

**Strengths And Weaknesses:**

**Strengths**
A new CL segmentation architecture, termed TILES, is proposed. The main strength of TILES lies in decomposing the learning process by conducting KD loss between the previous-task decoder's output and pseudo labels to update the encoder, and applying CE loss to the current-task decoder. Extensive experiments demonstrate the superiority of the proposed method.


**Weaknesses**
1. The main concern regarding the proposed method lies in its computational complexity in real-world applications. Although the experiments demonstrate that the proposed method has a relatively small number of parameters, it is worth noting that in practical applications, the number of object classes is often significantly larger than those found in public datasets. However, TILES is a method where the number of parameters grows linearly with the number of tasks, which raises a serious concern about its computational complexity when the number of classes exceeds thousands or even tens of thousands. Furthermore, every time a new class is received, the previous decoders need to be reused multiple times to infer pseudo labels and execute knowledge distillation loss, which means that the training budget is also severely affected when the number of categories is very large. Even though the authors use a lightweight model as the backbone, it still cannot fundamentally solve this problem. This actually contradicts the authors' claim, as they need to consider whether the proposed method can be applied to real-world scenarios, rather than just a few public datasets.


2. The authors have investigated the impact of different transformer-based backbones on performance. However, it is also interesting to explore the effect of different transformer-based segmentation networks on performance, such as Mask2Former and SegViT.

---

> ### Author Response · Authors · 2025-07-05
> **We thank the reviewer for its valuable comments. Changes in the main paper are highlighted in red.**
>
> 1:
>
> - In the absence of prior work using small backbones, the main goal of proposing TILES is to offer a first compelling alternative for scenarios with certain conditions: e.g., severe memory constraints and number of increments less than 239 steps for TILES-T or 43 steps for TILES-S (to not exceed the 102M parameters used by SOTA ViT-based methods). Note that this is the limit of the number of increments and each increment can have many classes. Indeed, the choice of architecture depends on the acceptable trade-off between performance and resource usage, as well as the maximum number of increments required by each application. Beyond 239 increments, we acknowledge that we cannot assert whether TILES-T or Incrementer (ViT-B) is the more effective choice. While scenarios with increments exceeding these values cause the augmentation of the memory footprint needed for TILES, we have no guarantee that Incrementer can be the better choice and whether it can handle catastrophic forgetting while updating both the encoder and the decoder parameters for hundreds or thousands of increments. A deeper investigation of this issue is out of the scope of our paper (which provides a first solution towards lightweight CI-SS under certain conditions) and is not currently feasible due to limitations in available academic CI-SS datasets. We encourage the research community for a deeper investigation of these use cases. More details have been added to the main paper for clarification.
>
> - In the training phase, every KD-based approach, including TILES, loads the model twice: the current and the previous model weights. Throughout the paper, we report the number of parameters used during inference for all methods. Note that we do consider the parameter expansion when computing the number of parameters for TILES. These values are doubled during training for all models due to the use of both current and previous models. We argue that while the expanding architecture of TILES can increase the memory usage, it remains limited for a relatively limited number of increments e.g. $+0.4M$ parameters for each increment for TILES-T, while providing significant performance gain compared to Incrementer (ViT-T).
>
> 2: Previous transformer-based CI-SS approaches provide such study as they are based on different segmentation algorithms: CoinSeg and MicroSeg are based on a Mask2Former+ DeepLabV3, RBC(ViT-B) and MiB(ViT-B) are based on a per-pixel classification model SETR, and Incrementer is based on Segmenter. While it is interesting to further study the impact of segmentation algorithms on the CI-SS methods, it is out of the scope of our paper where we focus on providing a suitable method for severe memory constraints use-cases while comparing with the current SOTA approach (Incrementer). We chose to retain the same semantic segmentation architecture to focus on the impact of the expanding components.

---

### Review · Reviewer_yy3t · 2025-06-22

**Summary Of Contributions:**

The authors explore class incremental learning in semantic segmentation architectures. They propose a new variant of incremental learning by adding capacity to the decoder network to mitigate forgetting effects. They evaluate this on different CI settings (pascal and ade20k) for semantic segmentation, and focus on comparisons while controlling the number of network parameters across different ViT based architectures. The key claim is that TILES is able to provide strong CI performance even on very small models; providing close to SOTA results with sufficiently less parameters on some of the considered datasets.

**Audience:**

Yes

**Claims And Evidence:**

No

**Requested Changes:**

Please see full list in "Strengths And Weaknesses", the key areas of improvement are:

1. Better discussion of the results, in particular the claim about SOTA performance with 14x less parameters needs to be discussed more nuanced. This is done later on in the results section, but neither in the abstract/intro. Right now, this result is either not visible from the way results are presented, or it is simply not the case in this generality and the claim needs to be downtuned.
2. Better presentation of results, in particular at least one plot should be added to underline the key claims of the paper; error bars should be added to this plot (currently not done in the tables, likely also due to space?)
3. Some statements in the paper need to be backed up by references and/or made more precise, see above.

Minor/typos:
- "i.e." is always missing the "." after the e; plus some uses of i.e. are off (e.g. in the "Knowledge remaining" section)
- In the ablation table, it might be more intuitive to put the "full setting" on top, and then show the variations of this?
- tables have a lot of vertical lines, which could be removed for a neater presentation

**Strengths And Weaknesses:**

The motivation for the work (memory-constrained evaluation in the IC setting) is generally acceptable, and the chosen experiments are reasonable (PASCAL and Ade20k), especially Ade20k. It seems that the considered settings are the ones typically employed for this kind of study. A chronolocial list of weaknesses and questions is listed below.

**Abstract**

- The first 6 lines of the abstract could be written a bit more concise.
- wrt -> write out
- "state-of-the-art" results reported in the abstract are overclaimed, given the experimental results

**1 Introduction**

- "fair": Implicitly, the authors claim that their notion of "fair" is the correct one and other experimental evaluations are incorrect in that regard. I would discuss this in a more nuanced way/clarify what is meant here.

**2 Related work**

- "Replay methods": These have been around for a long time, the earliest ref cited is from 2020. Would be good to discuss more of the earlier works in this field in continual learning, as replay has been used very extensively across contexts
- "In particular, knowledge distillation .... has shown great success reducing the castatrophic forgetting": Where was this reported? can you add a reference?
- "Therefore, we believe that transformer based arch can be more apprproirate for IC sicne they do not rely on BN": A clean way to test this would also be to use a convnet variant using groupnorm, layernorm or instance norm, and compare. Minimally, these methods should be discussed here, it is a limitation of the study to not include such architectures.
- "Moreover, a very important aspect of real world applications is the memory footprint": Since this is the statement the premise of the paper hinges on, it would be good to give concrete examples. What are these real world applications, and how big is the allowed memory footprint in these? It would be particularly good if later experiments could point out the actual footprint of the model on the GPU device to establish that these claims are relevant in real applications.

**3 Method**

- 3.2, "minimum forgetting thanks to their considerable number of parameters": reference needed
- Figure 1: is it possible to show more (non cherrypicked) examples of the segmentation between the different model sizes to get a qualitative feel of the metrics reported in the tables (ideally metrics are reported alongside different example images)

**Results**

- Table 1: I find the table very hard to read; what are the relevant comparison points for the claims made in the abstract? also given how many different settings you considered (which is good), a summary plot highlighting all settings and datasets would greatly help to bring the main message of the paper across, e.g. similar to Figure 1 in the EfficientNet paper (https://arxiv.org/pdf/1905.11946) might be suitable. (e.g. with #params on the x axis)
- Table 1: For e.g. 15-1, "all", the drop from SwinB to ViT-T is quite substantial (~71 to 45-55). While expected due to change in model size, the claim that model is SOTA except these 14x fewer parameters seems debatable given these results. On page 11 this is discussed more nuanced (limited to ADE20k, where it is good), but the claim above needs to be adapted to the actual results.
- Can the results marked with "+" which differ in mIoU be re-evalated so that all the numbers in the table are comparable? Right now this makes the results even harder to interpret. If it is challenging to recompute these numbers, please also provide this mIoU computations for the TILES models for comparison.
- "On the other hand, we can notice in the three tables" -> this would be a section that would benefit a lot from a summary figure, as suggested above, to clearly point out these general claims.
- "KR p.p." -> what does this stand for?
- Table 4 and 5: Can these be merged to make the parameter counts more readable and easier to relate? also, again, a summary plot is probably the even better way to discuss these results
- "good scaleability with the number of increments" -> max. increments seems to be 6? Could you discuss this claim more, why are there no experiments with larger number of increments?

---

> ### Author Response · Authors · 2025-07-05
> **We thank the reviewer for the valuable comments, including the minor requests for improvement and  correction of typos. We have carefully addressed all of them in the revised version of the manuscript. Changes in the main paper are highlighted in red for the reviewers' convenience.**
>
> Abstract: We have revised the abstract for improved clarity and conciseness. In particular, we agree that the final sentence could be misleading. It has been updated to: “TILES outperforms several previous methods on various challenging benchmarks while using up to 14 times fewer parameters.”
>
> Introduction: Additional clarification has been added regarding the fairness challenge.
>
> Related work:
>
> - Indeed, replay methods weren't discussed in details because they aren't considered in this work. Yet, we have added some earlier references, as requested.
>
> - We have added a reference to support the effectiveness of KD in CI-learning.
>
> - The fact that BN layers are inconvenient for continual learning is a claim by Li et al. (2022). As our work focuses primarily on introducing a lightweight ViT-based CI-SS approach, studying the effect of normalization in ConvNets seems beyond our current scope. We have added a comment about this.
>
> - Additional explanation regarding the importance of designing lightweight AI models has been included. Indeed, this is especially the case for edge computing environments, where optimizing resource usage for tasks like SS directly translates to more available computational power for other concurrent tasks, while preserving battery life. For instance, this is important for visual tasks on mobile phones, embedded systems in drones or autonomous robots relying on compact hardware.
>
> Method:
>
> - In general, using bigger backbones improves performance which has been proved in several non-incremental computer vision applications (EfficientNet, DINOv2[1]). Proving this aspect for CI-SS is studied in this paper and supported by experimental evidence. We changed the sentence to clarify that this isn't a previous claim needing reference.
>
> - To avoid overloading the overview figure, we chose not to include predictions from various backbones. Instead, we added Fig. 4, which represents segmentation outputs from Incrementer and TILES for two Pascal-VOC 15-1 overlapped samples. It illustrates that TILES maintains relatively strong performance with smaller backbones, whereas Incrementer exhibits a more significant performance drop, especially with ViT-T.
>
> Results:
>
> - For more clarity, we retain only the best Resnet-based approaches, and different versions of Incrementer and TILES in the 3 tables in the main article. These are sufficient to highlight the performance gains of ViT-based models and to compare TILES to Incrementer across backbones. A more exhaustive comparison is provided in Tab. 1, 3, and 5 of the appendix (A). As requested, we added an efficiency plot similar to the EfficientNet paper (cf. Fig. 1 of the new version of the main paper, with error bars). Yet, putting all scenarios in a single figure makes it illegible as points of different scenarios would overlap. Thus, we added separate plots per scenario in the A.
>
> - We made our claim clearer to avoid misunderstanding. Indeed, considering the absolute mIoU results, TILES doesn't outperform all previous SOTA, regardless of the backbone. However, TILES-B and/or TILES-S do outperform most previous methods: using up to 4 times fewer parameters, e.g., comparing SwinB approaches and TILES-S. Moreover, TILES-T outperforms most previous approaches while using up to 14 times fewer parameters on ADE20k. Now, considering the KR metric, TILES demonstrates great performance with its different versions.
>
> -  As stated by the reviewer, it's very challenging and sometimes impossible (missing code) to re-compute all methods to eliminate the background IoU from the all-mIoU value. As a compromise, we report the results of TILES considering the background IoU for Pascal-VOC scenarios in Tab. 2 and 4 of the A. Note that comparing methods considering homogeneous computations for both settings brings the same conclusions, as all methods succeed to segment well the background. In ADE20k annotation, the 0-labeled class doesn't carry semantic meaning, it only represents unlabeled pixels that should be omitted in the training process (Zhou et al. (2019)). TILES was trained to detect the 150 classes (1-150). No clear details about how other methods consider this class.
>
> - Summary figures have been added to the main article and the A as noted above.
>
> - KR p.p. stands for KR percentage points (added to the paper).
>
> - We think that merging Tab. 4 and 5 would make them very hard to read as they present completely distinct data. While the two tables seem to have the same lines, Tab. 4 represents the mIoU values for each backbone for the two datasets and Tab. 5 illustrates the number of parameters of each backbone for different numbers of incremental steps.
>
> - While different applications could suggest different settings of number of increments and number of classes per increment, we follow the CI-SS research community by considering the 9 mostly reported scenarios by SOTA methods.
>
> [1] DINOv2: Learning Robust Visual Features without Supervision, TMLR 2024.

---

> > ### Comment · Reviewer_yy3t · 2025-07-18
> >
> > Dear authors,
> >
> > thanks a lot for your rebuttal. Before entering my recommendation on this paper, I have two final clarification questions:
> >
> > Regardings metrics (IoU) computation: I understand it is complicated to re-evaluate the older methods using the "new" metric, but why is it challenging to report the performance of your model under the "old" IoU metric including the background?
> >
> > I still fail to see where the claim "TILES outperforms **several previous methods** on various challenging benchmark**s** while using up to 14 times fewer parameters" --- but maybe I am overlooking something. Could you clarify again the exact numbers you considered to back up this statement? The 14x has to refer to the difference between the larger (B) and the smallest (T) models. In Table 1, TILES-T outperforms MicroSeg+ in one of three settings and is on par with Bg_Adapt, but is worse than Incrementer-B and the two resnets in the remaining two (and note that the resnets are only about 7x larger, not 14x); in Table 2,  similar picture; in Table 3, the resnets are outperformed but the -B model just in one setting, one is on par, one is worse.
> >
> > This, I believe that this part in the abstract as-is still overclaims. It would be better to make a more accurate and nuanced statement, this won't hurt to the overall story of the paper. I think these gains are otherwise really good, just presented in a misleading way.

---

> > > ### Author Response · Authors · 2025-07-18
> > > **Clarification of two points**
> > >
> > > 1/ As stated in the rebuttal, we did provide the "old" IoU metric for all Pascal-Voc scenarios with the different versions of TILES in tables 2 and 4 in the Supplementary Material pdf file. We can see that when considering the background ("old" IoU metric), results for all-mIoU are consistently better:
> > > - On Pascal-VOC overlapped (resp. 19-1, 15-5, 15-1 scenarios):
> > > 	* TILES-B:
> > > 		- old all-mIoU (in \%): 80.3, 79.5, 74.5
> > > 		- new all-mIoU (in \%): 79.6,  78.8, 73.7
> > > 	* TILES-S:
> > > 		- old all-mIoU (in \%): 78.8, 76.9, 70.8
> > > 		- new all-mIoU (in \%): 77.8, 76.1, 69.8
> > > 	* TILES-T:
> > > 		- old all-mIoU (in \%): 72.7, 66.8, 56.7
> > > 		- new all-mIoU (in \%): 71.8, 65.6, 55.0
> > > - On Pascal-VOC disjoint (resp. 19-1, 15-5, 15-1 scenarios):
> > > 	* TILES-B:
> > > 		- old all-mIoU (in \%):  79.9, 71.3, 65.8
> > > 		- new all-mIoU (in \%): 79.2, 70.5, 64.5
> > > 	* TILES-S:
> > > 		- old all-mIoU (in \%): 78.5, 70.1, 65.4
> > > 		- new all-mIoU (in \%): 77.7,  69.0, 64.2
> > > 	* TILES-T:
> > > 		- old all-mIoU (in \%): 72.5, 62.2, 53.6
> > > 		- new all-mIoU (in \%): 71.5, 60.7, 51.7
> > > So we prefer to show the more challenging "new" metric in the paper, as it evaluates only semantically well-defined foreground segments.
> > >
> > > 2/ We say that "TILES outperforms several previous methods on various challenging benchmarks while using $\textbf{up}$ to 14 times fewer parameters", where 14x is the biggest ratio of parameters gain. We do confirm that this ratio is not always reached, but it is the upper bound. Moreover, for every studied scenario, TILES succeeds in  outperforming several of the previous approaches while using less parameters (ranging from 3x to 14x fewer):
> > > - In Pascal-VOC overlapped scenarios (performance values are all-mIoU in \%):
> > > 	* 19-1: TILES-T (71.8) outperforms UCD (70.0), SDR (67.4), and MiB (67.8) while using $\textbf{9x}$ fewer parameters (7.1M vs. 66M).
> > > 	* 15-5: TILES-S (76.1) outperforms RBC ViT-B (74.7), MiB ViT-B (74.7) and  MicroSeg+ ViT-B (75.2) using $\textbf{4x}$ fewer parameters (26M vs. 102M and 104M).
> > > 	* 15-1: TILES-T (55.0) outperforms UCD (41.9), SDR (39.2), PLOP (54.5) and MiB (29.7) using $\textbf{7x}$ fewer parameters (8.7M vs. 66M).
> > > - In Pascal-VOC disjoint scenarios (performance values are all-mIoU in \%):
> > > 	* 19-1: TILES-T (71.5) outperforms SDR (68.4) and MiB (67.6), and is on a par with UCD (71.5)  using $\textbf{9x}$ fewer parameters (7.1M vs. 66M).
> > > 	* 15-5: TILES-T (60.7) outperforms PLOP (59.8) using $\textbf{9x}$ fewer parameters (7.1M vs. 66M).
> > > 	* 15-1: TILES-T (51.7) outperforms RBC+ (51.1), UCD (42.9), SDR (48.1), PLOP (40.2) and MiB (37.9) using $\textbf{7x}$ fewer parameters (8.7M vs. 66M).
> > > - In ADE20k scenarios (performance values are all-mIoU in \%):
> > > 	* 100-50: TILES-T (37.0) outperforms CoinSeg+ SwinB (36.6) and MicroSeg+ SwinB (35.4) using $\textbf{14x}$ fewer parameters (7.1M vs. 104M). Moreover, TILES-S (44.8) outperforms all SOTA approaches while using up to $\textbf{4x}$ fewer parameters (26M vs. 104M).
> > > 	* 100-10: TILES-T (30.1) outperforms UCD (23.2), SDR (21.7), PLOP (22.3) and MiB (25.9) using $\textbf{7x}$ fewer parameters (8.7M vs. 66M). Moreover, TILES-S (36.4) outperforms all SwinB and ResNet-101 approaches while using up to $\textbf{3x}$ fewer parameters (33M vs. 104M).
> > > 	* 50-50: TILES-T (35.4) outperforms MicroSeg+ SwinB (32.5) using $\textbf{14x}$ fewer parameters (7.1M vs. 104M). Moreover, TILES-S (43.4) outperforms all SOTA approaches except Incrementer ViT-B (43.9) while using up to $\textbf{4x}$ fewer parameters (26M vs. 104M).
> > >
> > > Notice that TILES-S achieves particularly competitive results in all-mIoU on ADE20k dataset which involves 150 classes.
> > >
> > > To avoid misunderstanding, we will change this sentence to: "When using smaller backbones (typically, less than 33M parameters), TILES demonstrates performance gain compared to previous SOTA and even outperforms several previous approaches on various challenging benchmarks while using much fewer parameters".
> > >
> > > We will also add this analysis beside the complete tables of results in the Supplementary Material.

---

### Decision · Action_Editor_aqMG · 2025-08-12

**Recommendation:** Reject

**Additional Comments:**

The article received split recommendations, with yy3t being slightly positive (while still finding that performance claims needed to be down-toned) while the others (tUFz, rhM8) being negative, with concerns on the study of the proposed technical components (tUFz) and limits on the experimental evaluation, i.e., setting, backbones (rhM8). The AE went over the manuscript and agreed with some of the concerns of the reviewers (highlighted above), thus the reject recommendation for the current version of the manuscript.

Nevertheless, the AE agrees with yy3t regarding the value of the task and that the approach is interesting. With revised claims and expanded studies, the article would be a stronger contribution to the field (even in case of some negative results):  authors may consider submitting a major revision of the manuscript to this venue.

**Audience:**

Yes

**Audience Explanation:**

The article treats the problem of class-incremental learning in semantic segmentation. The manuscript would be of interest to the community working on this specific computer vision problem as well as researchers in continual learning.

**Claims And Evidence:**

No

**Claims Explanation:**

The article presents a method, TILES, whose purpose is to perform continual learning in semantic segmentation in memory-constrained settings. The core claims are that (i) the paper provides a fair comparison across different CI-SS methods, demonstrating their performance drop for smaller backbones; (ii) TILES is the best in solving CI-SS with critical memory constraints. While these contributions have merits, some claims are not fully reflected in the manuscript. Specifically, only Incrementer is tested with the same number of parameters of TILES, thus the comparison/study is limited to this approach. Moreover, while the performance gain w.r.t Incrementer is notable, the claim that the method achieves better results with up to 14x reduction in parameters is quite stretched as this rarely happens, e.g., only in one out of 6 settings (i.e., Tab. 1, Pascal-VOC overlapped, TILES-T vs the best SWIN-B based model). Moreover, reviewers found the proposed technical components to not have been sufficiently studied  Overall, while the article contains interesting insights, some claims need to be revised and additional evidence be included.

**Resubmission Of Major Revision:**

The authors may consider submitting a major revision at a later time.